# Microbiome composition and function within the Kellet's whelk perivitelline fluid

Benjamin N. Daniels,[1] Jenna Nurge,[1] Chanel De Smet,[1] Olivia Sleeper,[1] Crow White,[1] Jean M. Davidson,[1] Pat Fidopiastis[1]

**ABSTRACT** Microbiomes have gained significant attention in ecological research, owing to their diverse interactions and essential roles within different organismal ecosystems. Microorganisms, such as bacteria, archaea, and viruses, have profound impact on host health, influencing digestion, metabolism, immune function, tissue development, and behavior. This study investigates the microbiome diversity and function of Kellet's whelk (*Kelletia kelletii*) perivitelline fluid (PVF), which sustains thousands of developing *K. kelletii* embryos within a polysaccharide and protein matrix. Our core microbiome analysis reveals a diverse range of bacteria, with the *Roseobacter* genus being the most abundant. Additionally, genes related to host-microbe interactions, symbiosis, and quorum sensing were detected, indicating a potential symbiotic relationship between the microbiome and Kellet's whelk embryos. Furthermore, the microbiome exhibits gene expression related to antibiotic biosynthesis, suggesting a defensive role against pathogenic bacteria and potential discovery of novel antibiotics. Overall, this study sheds light on the microbiome's role in Kellet's whelk development, emphasizing the significance of host-microbe interactions in vulnerable life history stages. To our knowledge, ours is the first study to use 16S sequencing coupled with RNA sequencing (RNA-seq) to profile the microbiome of an invertebrate PVF.

**IMPORTANCE** This study provides novel insight to an encapsulated system with strong evidence of symbiosis between the microbial inhabitants and developing host embryos. The Kellet's whelk perivitelline fluid (PVF) contains microbial organisms of interest that may be providing symbiotic functions and potential antimicrobial properties during this vulnerable life history stage. This study, the first to utilize a comprehensive approach to investigating Kellet's whelk PVF microbiome, couples 16S rRNA gene long-read sequencing with RNA-seq. This research contributes to and expands our knowledge on the roles of beneficial host-associated microbes.

**KEYWORDS** microbiome, RNA-seq, antimicrobial activity, antibiotic biosynthesis, gene expression, GO term, microbial communities

Communities of microorganisms living in a habitat within a host organism, or microbiomes (1), can exhibit complex and significant interactions with their host, influencing the host's digestion, metabolism, immune function, and behavior. For example, some microbes contain antimicrobial properties that help the host defend against foreign invaders, such as antimicrobial compounds from dimethylsulfoniopropionate (DMSP)-metabolizing bacteria and the production of bacteriocins (2–4), and present specific metabolites that support host growth (5, 6).

The diversity of microbes can be highly variable, differing among host individuals of a species in relation to the individual's diet, surrounding environmental conditions (e.g., water temperature), captivity, geographic location, and other factors (7–12). Thus,

Address correspondence to Benjamin N. Daniels, ben.daniels255@gmail.com.

The authors declare no conflict of interest.

See the funding table on p. 17.

in relation to these factors, differences in the composition and function of microbiomes may be evident between individuals and populations of the host species.

In this study, we explored the diversity and functional role of the microbiome in the perivitelline fluid (PVF) in egg capsules of Kellet's whelk, *Kelletia kelletii*, a coastal marine, predatory gastropod, and commercial fisheries species along the North American West Coast (13). Similar to some other marine invertebrates [e.g., near-shore squid (*Doryteuthis opalescens*), Apple Snail (*Pomacea maculata*), and Horseshoe crab (*Tachypleus tridentatus*)], Kellet's whelk exhibits a bi-partite life history (dispersive larvae, relatively sessile adults) and females lay egg capsules on hard substrate (e.g., reef) in the coastal marine environment, each containing hundreds to potentially more than 1,000 eggs that develop (prior to dispersal) for 35–45 days while suspended in a PVF consisting of a polysaccharide and protein matrix (14–17). This PVF most likely originates from an albumen exocrine gland within the mother (18). The PVF secreted by these glands have been reported to nourish and protect developing eggs. For example, in Zebrafish (*Danio rerio*) and the sea hare (*Aplysia kurodai*), the PVF contains immune defense proteins and antibacterial factors that protect embryos from pathogenic infection (15, 19), and the PVF of the Apple Snail produces plant-like defenses that deter predation and pathogenic infection (20–23); for both species, the defense mechanisms are believed to be maternally inherited. The PVF may also contain functional microbiomes that can be vertically transferred to the PVF of egg capsules. Little is known about the diversity and function of microbial communities in relation to marine invertebrates, especially in PVF, due to their underrepresentation in scientific research (24). However, existing literature suggests that microorganisms may be deposited into the PVF and on the surface of eggs by egg-laying mothers in order to contribute to the embryo development and protection (25–27). More specifically, the accessory nidamental gland (ANG) in the Hawaiian bobtail squid (*Euprymna scolopes*) deposits the egg jelly coat along with symbiotic bacteria that prevent egg fouling. The ANG is a similar gland to the albumen exocrine gland (28, 29). Likewise, Vesicomyid clams rely on chemosynthetic bacteria for harnessing chemical energy and are vertically transmitted from the mother (30). These findings support the investigation of the microbial community in Kellet's whelk's PVF and its contribution to protecting and nurturing the gastropod's developing embryo. Further research is required to identify the origin of these microbes and whether they are a result of a maternally packaged inoculate or directly recruited from the environment.

Kellet's whelk recently exhibited a northern range expansion beyond its historical range in Baja California, Mexico, and the Southern California Bight, USA, to colder-water habitat along the central California coast, USA (31), potentially in response to climate change (32). The contrasting temperatures between these historical and expanded range regions correspond with distinct differences in the diversity and composition of microbiomes of fish and invertebrates (33, 34), as well as the parasites that associate with them. For example, Kellet's whelk individuals from the expanded range experience substantially lower parasite diversity and a different composition of parasites than historical-range individuals (35). Given these observations and that the microbial community in Kellet's whelk PVF may be delivered by the mother, we hypothesized differences in microbiome composition and function in PVF egg capsules laid in a common environment by Kellet's whelk collected over a broad geographic range.

To characterize and compare the composition of the Kellet's whelk PVF microbiomes, we conducted 16S rRNA gene long-read sequencing on egg capsules with parents originating from four different locations in California: Monterey (MON), Diablo Canyon (DIC), Naples (NAP), and Point Loma (POL). The results of this study characterize the microbiome associated with each of these locations and the common microbiome within Kellet's whelk PVF. A subset of samples was subjected to sequencing using archaea-specific primers to gain insight into the diversity and function of archaea within Kellet's whelk PVF. Total mRNA sequencing results from 58 samples with parents originating from three of these locations (MON, NAP, and POL) were leveraged to identify gene expression of the PVF microbiome. This study identifies the potential defense

mechanisms within Kellet's whelk PVF provided by the symbiotic microbes through RNA-seq transcriptome assembly and analysis. Understanding the microbial diversity and function during this important life history trait will give greater insight into the importance and function of symbiotic relationships among hosts and microbes.

## MATERIALS AND METHODS

### Common garden experiment

Adult Kellet's whelk was collected from sub-tidal reefs located near Monterey (MON; 36.6181670 N, 121.897 W), Diablo Canyon (DIC; 35.2244500 N, 120.877483 W) Naples (NAP; 34.4219670 N, 119.952283 W), and Point Loma (POL; 32.665333 N, 117.261517 W), California, in 2019, transported live to the aquaria at the California Polytechnic State University research pier, located in Avila Beach, California, USA, and maintained under ambient conditions with food in the form of frozen seafood provided *ad libitum* (CDFW Scientific Collection Permit 8018 to C.W.) (Fig. 1). Two sampling sites were located in the species' expanded range (MON and DIC) and two from the historic range (NAP and POL). Adults were given a full year to adjust to new conditions before egg capsule samples

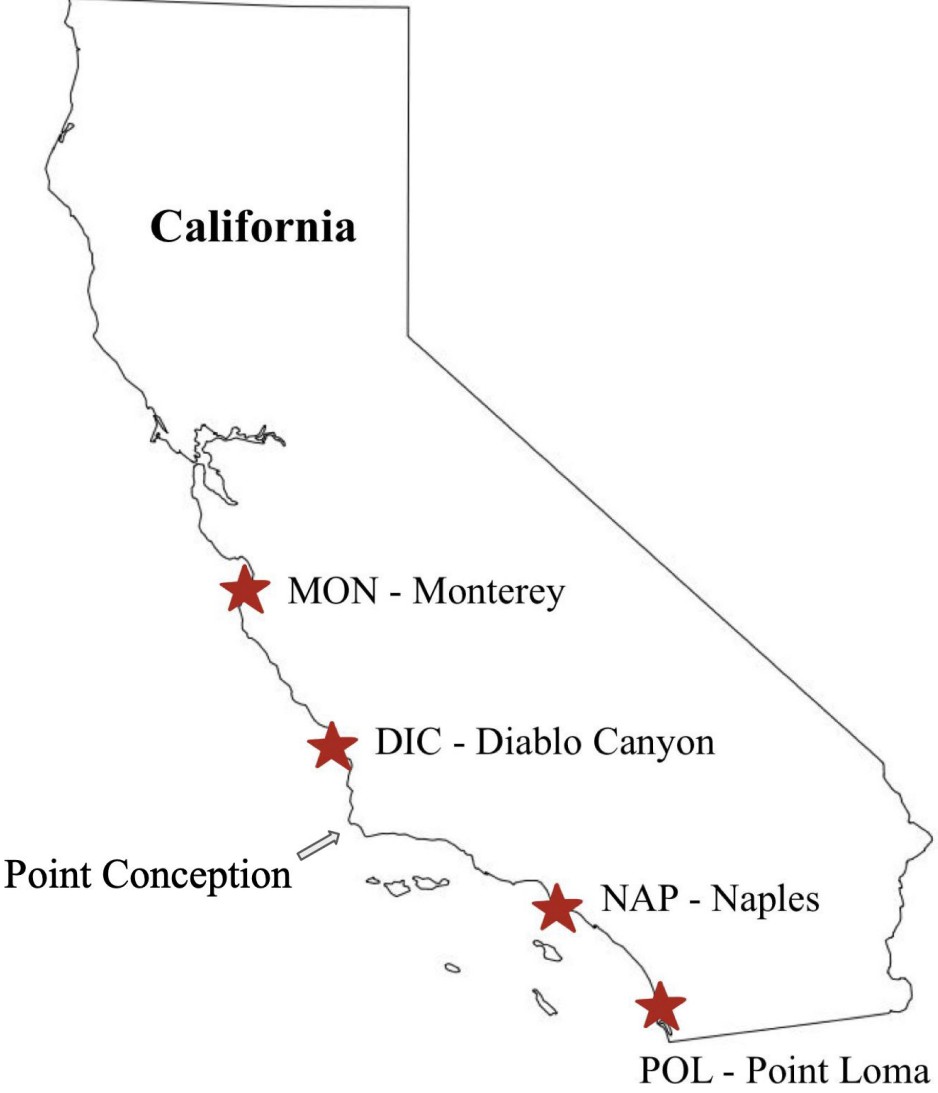

**FIG 1** Collection sites of Kellet's whelk used in this experiment. Labeled codes MON, DIC, NAP, and POL correlate to the nearby cities for simplicity. MON and DIC represent a recent range expansion north of Pt. Conception into colder-water habitat (31).

were collected in 2020. Egg capsules were collected at peak development (35–45 days after laying), flash frozen in liquid nitrogen, and stored at −80°C. For each location, four egg capsules, each from a different brood, were prepared for DNA extractions. A total of 16 egg capsules were collected for DNA extractions. Egg capsules were selected from different broods to represent the microbiomes from different mothers of each location. To ensure that the eggs in each of our samples were at the same developmental stage, eggs from the same brood were observed under a microscope and their development stage was identified. A total of 58 egg capsules (24 samples from NAP, 18 samples from MON, 6 samples from POL, and 10 samples from MON x NAP; potential crossings between the two locations) were prepared for RNA sequencing by removal of the parental egg capsule. Samples were flash frozen in liquid nitrogen and sent to Novogene (University of California, Davis, CA, USA) for RNA extractions and sequencing.

## DNA extraction and 16S sequencing

DNA extractions of four egg capsules from laboratory-maintained animals originating from each location (MON, DIC, NAP, and POL) were conducted using the DNeasy PowerSoil Pro kit from Qiagen (QIAGEN, Aarhus, Denmark). Egg capsule lining was removed from each sample, and DNA extractions were conducted according to the manufacturer's instructions with slight modification to the cell lysis step. Briefly, ~30 mg of PVF was mixed with cell lysis components, subjected to mechanical homogenization using a tissue grinder, followed by DNA extraction. Eluted DNA was stored at −20°C. Further details on the extraction method can be found here.

All sample DNA concentrations were measured by fluorescence using a BR DNA Qubit Assay (Invitrogen, MA, USA), and quality was measured by gel electrophoresis in a 1% gel with ~250 ng of DNA per sample at 80 V for 45 minutes (Fig. S1). For bacterial diversity and abundance measurements, the three highest quality DNA samples from each location (12 total) were sent to MR DNA (Texas, USA) for long-read sequencing of the V1–V9 region of the 16S gene (Table S2). First, the 16S rRNA gene PCR primers 27F (5′-AGAGTTTGATCCTGGCTCAG-3′) and 1492R (5′-TACGGYTACCTTGTTACGACTT-3′) were used with each sample in a 35-cycle PCR. PCR was conducted using the HotStarTaq Plus Master Mix Kit made by Qiagen. Samples were cycled at 94°C for 3 minutes, 35 cycles of 94°C for 30 seconds, 53°C for 40 seconds, 72°C for 90 seconds, and 72°C for 5 minutes. The PCR product was checked with a 2% agarose gel electrophoresis for band intensity. The PCR pool was purified using Ampure PB beads (Pacific Biosciences, CA, USA), and libraries were prepared. DNA libraries were sequenced at MR DNA (www.mrdnalab.com, Shallowater, TX, USA) using the PacBio Sequel following the manufacturer's guidelines. Raw sequence data were further filtered for Circular Consensus Sequencing using Pacific Biosciences' algorithm. For archaeal diversity and abundance measurements, four NAP DNA samples were sent for 16S rRNA gene sequencing through MR DNA's diversity assay bTEFAP Illumina 20 k inhouse arc349F (5′-GYGCASCAGKCGMGAAW-3′). The sequence data were then processed using the MR DNA analysis pipeline. Briefly, operational taxonomic units (OTUs) were defined by clustering at 97% similarity. To classify final OTUs taxonomically, BLASTn was used against a 16S database from NCBI curated and regularly updated by MR DNA (www.ncbi.nlm.nih.gov).

## 16S data analysis

The data output provided by MR DNA included both the raw data sequences and the percentages and counts of bacterial and archaeal genus, species, identity, unknown, and OTU for each sample. Because the sequencing conducted was 10 k long-read sequencing of the entire 16S rRNA gene, the specificity of the bacterial identification was accurate to the genus and species level. As noted, however, this was not a quantitative assay and the values from each organism (bacteria) abundance were relative to the total number of organisms within each sample. For identifying the shared microbiome between all samples and locations, an OTU was considered to be present if two of the three samples within that location contained a percent abundance > 0. To obtain the relative percent

abundance for each location, the percent abundances of each OTU were averaged among the three samples. Similarly, to determine the overall relative percent abundance of the OTU within the core microbiome, the percent abundances of each OTU were averaged across all locations. Microbes that were common to all four locations were considered the core microbiome. The core microbiome phylogeny was constructed using the species identified in PhyloT and the NCBI taxonomy database (36). The phylogeny was constructed using the default Newick format tree file.

## RNA extraction and sequencing

RNA extractions, sequencing, and data quality control was conducted by Novogene. Egg capsule lining was removed from each sample, and RNA extractions were conducted. Briefly, RNA was extracted using TRIzol and RNA phase separation. After fragmentation, the first-strand cDNA was synthesized using random hexamer primers, followed by the second-strand cDNA synthesis using dTTP for non-directional library preparation. Messenger RNA (mRNA) was purified from total RNA using poly-T oligo-attached magnetic beads. Libraries were prepared using end repair, A-tailing, adapter ligation, size selection, amplification, and purification. Libraries were checked using Qubit and real-time PCR for quantification and bioanalyzer for size distribution detection. Libraries were pooled and sequenced on the Novaseq (Illumina, San Diego, CA). Raw data FASTQ files were processed through in-house Perl scripts. Clean reads were obtained by removing reads containing adapters, reads containing poly-N, and low-quality reads. Q20, Q30, and GC contents were calculated for the clean reads. RNA-seq reads were further cleaned using Trimmomatic v0.39 IlluminaClip with leading:3 trailing:3, slidingwindow: 4:15, minlen:36 parameters (SI 1). All downstream analyses were conducted using the clean data.

## Microbial transcriptome assemblies and function

The trimmed and paired RNA reads from all 58 samples were assembled using RNAspades v3.15.4 and the default parameters. RNAspades transcriptome was further filtered by removal of contigs without BLASTx hits to either the UniProtKB/Swiss-Prot database or the NCBI NR database (v2022_07) (37). To distinguish whether contig sequences were originating from host (Kellet's whelk) or microbial members (bacteria, viruses, and archaea), each contig was aligned to UniProt databases of Eukaryota (representing Kellet's whelk gene expression) or bacteria, viruses, and archaea (representing the microbial gene expression). Protein databases for eukaryotes, bacteria, viruses, and archaea were created using the Swiss-Prot annotated genes for each group using BLASTx (V2.12.0). The transcriptomes was aligned to each database and given their group identifier, i.e., BLASTx results for Eukaryota were given "host" in the identifier column and BLASTx results for microbes were given "microorganism" in the identifier column. All contig BLASTx results were ordered by bitscore, and duplicate contig IDs were removed. This resulted in each contig being assigned to its highest BLASTx bitscore and a group identifier. The contig IDs from the "microorganism" group were assembled into a microbial transcriptome. Gene count matrices for each contig were created using Kallisto (V0.48.0) across the 42 samples associated with parents from one location (NAP, POL, and MON) rather than crossings. Gene expression data were normalized between samples using transcripts per million (TPM). Gene Ontology (GO) terms were extracted from each transcriptome BLAST results and separated into molecular functions, biological processes, and cellular components. Expression data were further filtered based on associated GO terms and their log(TPM) as well as keywords in their functional annotation. The GO term bar graph was plotted using https://www.bioinformatics.com.cn/en, a free online platform for data analysis and visualization, and GO term similarity was visualized using REVIGO (38). See supplementary code for commands and scripts used.

## Differential expression

Differential expression between the main North (MON) and South (NAP and POL) populations was conducted using R Statistical Software (v4.2.2) (39) using DEseq2 (40). The Kallisto gene count matrices were used along with the MetaData (if the sample belonged to the North or South) to run DEseq2. Principal component analysis plots were created to show clustering of samples based on microbial gene expression. The adjusted *P* value (padj) was used to determine whether the gene was differentially expressed or not, based on a statistically significant padj threshold of 0.05. See supplementary code file for commands and scripts used.

## RESULTS

### Microbiome diversity

#### Location comparison

From each of the four locations, there were a total of 32,209 OTUs obtained from MON samples, 28,914 OTUs obtained from DIC samples, 35,934 OTUs obtained from NAP samples, and 33,818 OTUs obtained from POL samples.

For all four sites, the Proteobacteria were the dominant phyla. Bacteroidota were typically the second most abundant other than for MON (Table S1). Analysis of each sample location at lower taxonomic levels revealed clear dominance of two genera: *Roseobacter* and *Planktomarina* across all samples (Fig. 2 and 3A). Some genera were consistently found across all samples but were variable in their abundance including

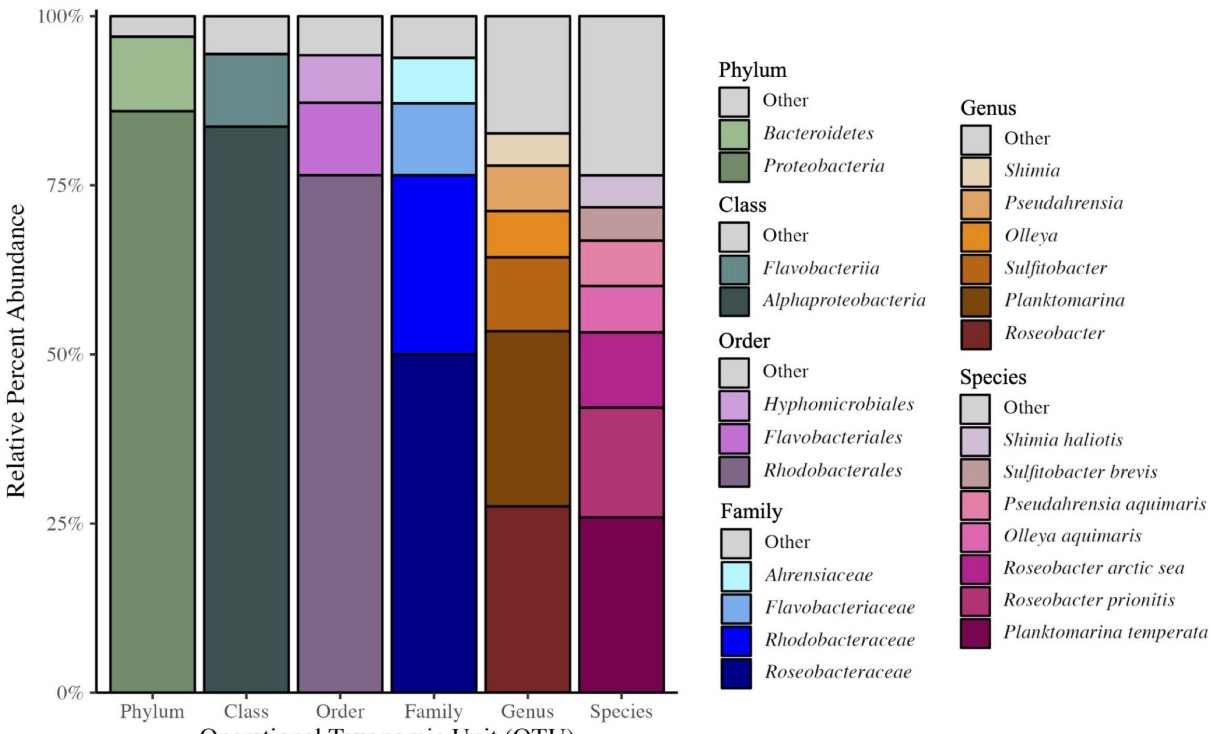

**FIG 2** Phylogenetic breakdown based on OTU of 16S DNA sequencing of the core microbiome. Eight organisms were found in the core microbiome to contribute greater than 5% of the relative abundance. To see the full list of core microbiome species, refer to Table S3. Phylum: Other 3.0%, Bacteroidetes 11.0%, Proteobacteria 85.9%; Order: Other 5.6%, Flavobacteriia 10.7%, Alphaproteobacteria 83.7%; Class: Other 5.8%, Hyphomicrobiales 7.0%, Flavobacteriales 10.7%, and Rhodobacterales 76.5%; Family: Other 6.2%, *Ahrensiac*eae 6.7%, *Flavobacteriaceae* 10.6%, *Rhodobacteraceae* 26.5%, and *Roseobacteraceae* 50.0%; Genus: Other 17.4%, *Shimia* 4.8%, *Pseudahrensia* 6.7%, *Olleya* 6.9%, *Sulfitobacter* 10.9%, *Planktomarina* 25.9%, and *Roseobacter* 27.5%; and Species: Other 23.5%, *Shimia haliotis* 4.7%, *Sulfitobacter brevis* 4.9%, *Pseudahrensia aquimaris* 6.7%, *Olleya aquimaris* 6.8%, *Roseobacter arctic sea* 11.2%, *Roseobacter prionitis* 16.2%, and *Planktomarina temperata* 25.9%.

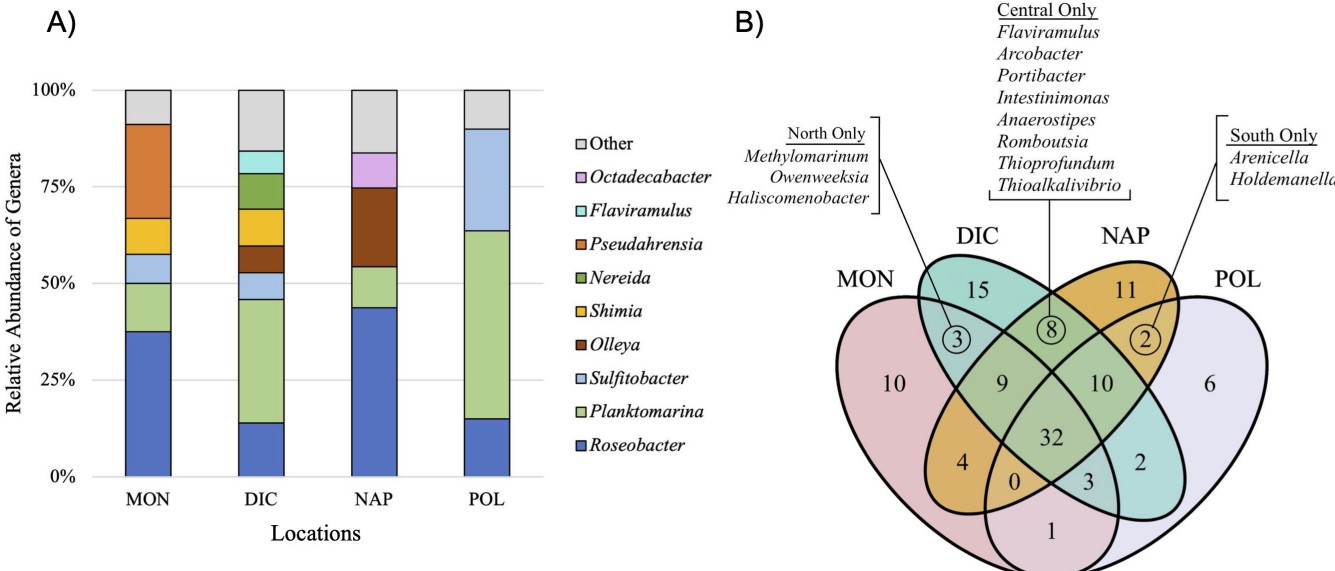

FIG 3 (A) Relative distribution of genera within each location. Only genera that were present and greater than 5% are represented. The genera that made up less than 5% are included in the "Other" block. (B) Venn diagram representing genera OTUs for all locations. The separate and overlapped sectors signify the unique and shared OTUs. The North, Central, and South headings represent the genera found that were unique to those locations along the coast.

*Sulfitobacter*, *Olleya*, *Shimia*, *Nereida*, and *Pseudahrensia* (Fig. 3A). MON contained 63 different genera and is the second lowest in diversity in comparison to the other locations (Fig. S2). *Roseobacter* dominated in relative abundance within MON at 37.5%, and, unlike other locations, *Pseudahrensia* contained roughly 24.3% relative abundance whereas the other locations contained less than 2% of *Pseudahrensia*. POL displayed the lowest amount of diversity within its microbiome and was primarily dominated by genera belonging to *Planktomarina* (48.6%), *Sulfitobacter* (26.3%), and *Roseobacter* (15.0%) (Table 1). The remainder of genera found within POL made up less than 1% of the microbiome. NAP, similar to MON, was dominated by *Roseobacter* (37.48%) and the second highest in diversity. DIC had the highest diversity with 83 different genera as well as the most even distribution of bacterial abundance. Similar to POL, DIC was primarily dominated by *Planktomarina* (31.88%) (Table 1).

TABLE 1  Ranking of the top 12 abundant genera within the samples MON, DIC, NAP, and POL based on the average relative percentage of each genus

| Rank | Genus | MON % abundance | DIC % abundance | NAP % abundance | POL % abundance |
|---|---|---|---|---|---|
| 1 | *Roseobacter* | 37.48 | 13.93 | 43.68 | 15.02 |
| 2 | *Planktomarina* | 12.54 | 31.87 | 10.59 | 48.59 |
| 3 | *Sulfitobacter* | 7.52 | 6.95 | 2.87 | 26.29 |
| 4 | *Olleya* | 0.01 | 6.87 | 20.48 | 0.05 |
| 5 | *Pseudahrensia* | 24.33 | 1.27 | 0.84 | 0.42 |
| 6 | *Shimia* | 9.26 | 9.62 | 0.04 | 0.10 |
| 7 | *Nereida* | 1.03 | 9.21 | 0.81 | 0.28 |
| 8 | *Octadecabacter* | 0.17 | 0.22 | 8.97 | 0.10 |
| 9 | *Flaviramulus* | 0.00 | 5.77 | 0.53 | 0.01 |
| 10 | *Gaetbulibacter* | 0.00 | 0.01 | 0.00 | 4.98 |
| 11 | *Ardenticatena* | 0.26 | 2.68 | 0.22 | 0.30 |
| 12 | *Polaribacter* | 0.02 | 1.21 | 1.21 | 0.00 |
| 13 | *Amylibacter* | 0.24 | 1.15 | 0.52 | 0.22 |
| 14 | *Pseudomonas* | 0.06 | 1.76 | 0.04 | 0.03 |
|  | Other | 3.89 | 3.42 | 4.95 | 1.76 |

We found no statistical significance between the Shannon diversity index of North vs South populations or between each location (Fig. S9; Table S9). Moreover, we found no significant relationship between microbial community dissimilarity when investigating beta diversity (Bray-Curtis distances) between locations and clustering into North and South groups (Fig. 4B). High variability was found within sample locations for each genus, especially at higher percent abundances (Fig. S8). Comparison of shared presence vs absence conducted between all locations revealed genera unique to and shared between each location (Fig. 3B). DIC location had the most unique genera (15) while POL had the least (6). Only three genera were unique to the two North locations (MON and DIC), and only two genera were unique to the two South locations (NAP and POL). Interestingly, the two central locations (DIC and NAP) had eight unique genera (Fig. 3B).

## Core microbiome

The core microbiome revealed the major contributors to Kellet's whelk PVF microbiome from the phylum to species level (Fig. 2; Table S3). Six genera were found to contribute approximately 82.6% of the core microbiome, with *Roseobacter* and *Plantomarina* representing 27.5% and 25.9% relative abundance, respectively. The subsequent four genera included *Sulfitobacter* with 10.9%, *Olleya* with 6.9%, *Pseudahrensia* with 6.7%, and *Shimia* with 4.8% relative abundance. The remaining 17.4%, depicted in the "Others" category in Fig. 2, consisted of 26 genera. The next two most abundant genera within the "Others" category were *Nereida* with 2.8% and *Octadecabacter* with 2.4% relative abundance (Table S3). The remainder of genera constituted less than 1% of the core microbiome and can be found in Table S3. Although precautions were taken to prevent contamination, some of the less abundant core microbiome species may also be a result of contaminants from the common garden environment.

Due to the high accuracy of the 16S rRNA V1–V9 long-read sequencing conducted, we were able to identify OTUs to the species level. There were 32 species identified to be shared with all locations (Table S3). Of those 32, 11 had abundances above 1% including *Planktomarina temperata* (25.89%), *Roseobacter prionitis* (16.20%), *Roseobacter arctic sea* (11.10%), *Olleya aquimaris* (6.83%), *Pseudahrensia aquimaris* (6.71%), *Sulfitobacter brevis* (4.94%), *Shimia haliotis* (4.7%), *Sulfitobacter pontiacus* (3.90%), *Nereida ignava* (2.8%), *Octadecabacter orientus* (2.20%), and *Sulfitobacter guttiformis* (1.50%) (Table S3). The phylogeny of bacterial species identified to be within the core microbiome indicates a wide variety of taxonomic families present with *Roseobacteraceae* being the most diverse (Fig. S5).

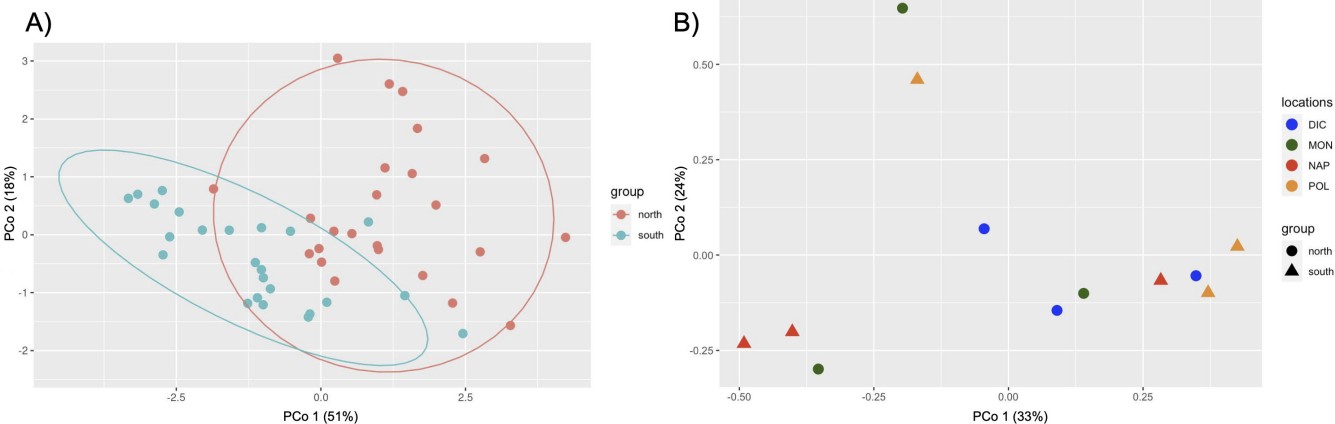

**FIG 4**  (A) Principal component analysis plot showing clusters of samples to visualize the differential expression of 22 genes between North and South locations (padj < 0.05). (B) Principal component analysis of beta diversity using Bray-Curtis dissimilarity with each sample plotted. Each location is represented by a different color and North locations are indicated by circle points while South locations are indicated by triangle points.

## Archaeal diversity

NAP was the only location sampled for archaea. A total of 8,754 OTUs were identified and mapped to 16 known archaeal OTUs with greater than 97% homology. The abundance of each OTU in each sample is shown in Fig. S3. Of the identified archaeal species, *Candidatus giganthauma insulaporcus* dominated the abundance of archaeal OTUs across all samples with an average percent abundance of 72%. Other archaeal OTUs identified were *Nitrosopumilus* spp., *Methanococcus voltae*, *Methanosaeta* sp., *Methanosaeta concilii*, and *Methanobacterium* sp. (Fig. S4). The archaeal gene expression was filtered using GO terms, and GO term similarity was mapped in Fig. S7.

## Viral diversity

Of the contigs mapped to the viral transcriptome, six viruses were identified with percent identity > 97%. Of these six identified viruses, three were most closely related to the following bacteriophages: *Escherichia* phage lambda, *Escherichia* phage T7, and *Enterobacteria* phage 434. Other gene expression was also identified with BLAST hits originating from 28 different bacteriophages (Bitscore > 40, *E*-value < 0.00001) (Table S8).

## Differential expression

Overall, there was limited to no significant differential expression of genes between the North and South locations for bacteria, viruses, and archaea with only 0.16% of genes being differentially expressed (Table S4; Fig. 4A). For bacteria, out of the 11,561 genes found to originate from bacteria in the assembled microbial transcriptome, there were 19 genes found to be differentially expressed between North and South locations. For archaea, there were no genes found to be differentially expressed out of the total 301 archaeal genes. Viruses had three differentially expressed genes out of 2,281 total genes (Table 2). A correlation test was conducted to compare each sample's gene expression and revealed more similar gene expression between NAP and MON with intralocation variation of expression for POL (Fig. S6).

**TABLE 2** Differentially expressed genes and their associated protein expression from bacteria and viruses between North and South populations

| Gene product | padj | Origin |
|---|---|---|
| Endoglucanase F (EGF) (cellulase F) (endo-1,4-beta-glucanase) | $5.13E^{-09}$ | Bacteria |
| Virulence metalloprotease (vibriolysin) | $7.68E^{-09}$ | Bacteria |
| Putative ankyrin repeat protein RF_0381 | $1.21E^{-08}$ | Bacteria |
| L-Glyceraldehyde 3-phosphate reductase (GAP reductase) | $1.19E^{-07}$ | Bacteria |
| Replicase polyprotein 1 a (pp1a) (ORF1a polyprotein) | $5.44E^{-07}$ | Viruses |
| L-Arginine-binding protein | $6.68E^{-07}$ | Bacteria |
| Type I restriction enzyme MpnII specificity subunit (S protein) (S.MpnII) | $4.53E^{-06}$ | Bacteria |
| Glyoxylate/hydroxypyruvate reductase A | $4.22E^{-05}$ | Bacteria |
| Putative glycosyltransferase HI_0258 | $6.26E^{-05}$ | Bacteria |
| DNA polymerase III PolC-type (PolIII) | $1.02E^{-04}$ | Bacteria |
| Uncharacterized protein LF3 | $2.49E^{-04}$ | Viruses |
| Neutral protease (aeromonolysin) (vibriolysin) | $3.61E^{-04}$ | Bacteria |
| Probable hemoglobin and hemoglobin-haptoglobin-binding protein 3 | $5.17E^{-04}$ | Bacteria |
| Expansin-YoaJ (EXLX1) | $5.39E^{-04}$ | Bacteria |
| Endoglucanase B (cellulase B) (endo-1,4-beta-glucanase B) | $1.11E^{-03}$ | Bacteria |
| Cys-loop ligand-gated ion channel (ELIC) | $1.51E^{-03}$ | Bacteria |
| Probable hemoglobin and hemoglobin-haptoglobin-binding protein 3 | $1.82E^{-03}$ | Bacteria |
| Gag-Pol polyprotein | $3.97E^{-03}$ | Viruses |
| N-Acetylmannosamine 1-dehydrogenase (NAM-DH) | $5.04E^{-03}$ | Bacteria |
| Probable hemoglobin and hemoglobin-haptoglobin-binding protein 3 | $1.03E^{-02}$ | Bacteria |
| Anti-sigma-I factor RsgI6 (endo-1,4-beta-xylanase) | $1.08E^{-02}$ | Bacteria |
| Probable hemoglobin and hemoglobin-haptoglobin-binding protein 3 | $2.78E^{-02}$ | Bacteria |

The three differentially expressed viral genes either were uncharacterized or were proteins with many different cleavage protein isoforms (pp1a and Gag polyprotein) and may be capable of possessing many different functions (Table 2). Of the 19 differentially expressed bacterial genes, four are related to host hemoglobin binding for heme uptake, three are related to cellulose digestion, and the others have ranging known or unknown functions such as methylation, binding affinity, and enzyme activity (Table 2).

## Microbiome function

To discover microbiome activity and function in our system, we investigated the transcriptomic data produced from all samples in the common garden experiment described above. A total of 14,143 genes were identified in the microbial transcriptome with 11,561 genes originating from bacteria, 2,281 genes originating from viruses, and 301 genes originating from archaea. Of the total number transcripts and genes sequenced, 10.39% and 4.19% originated from the microbiome, respectively. The activity and function of the microbial transcriptome in relation to the GO terms [biological process (BP), molecular function (MF), and cellular component (CC)] became evident through cross referencing with gene expression data of each sample. GO terms such as "siderophore transmembrane transport," "proteolysis," and "metal ion binding," were identified as highly active (sums of the gene expression associated with these GO terms were among the highest) GO terms within the microbiome (Fig. 5). GO term similarity for the top 40 expressed GO terms is shown in Fig. 6.

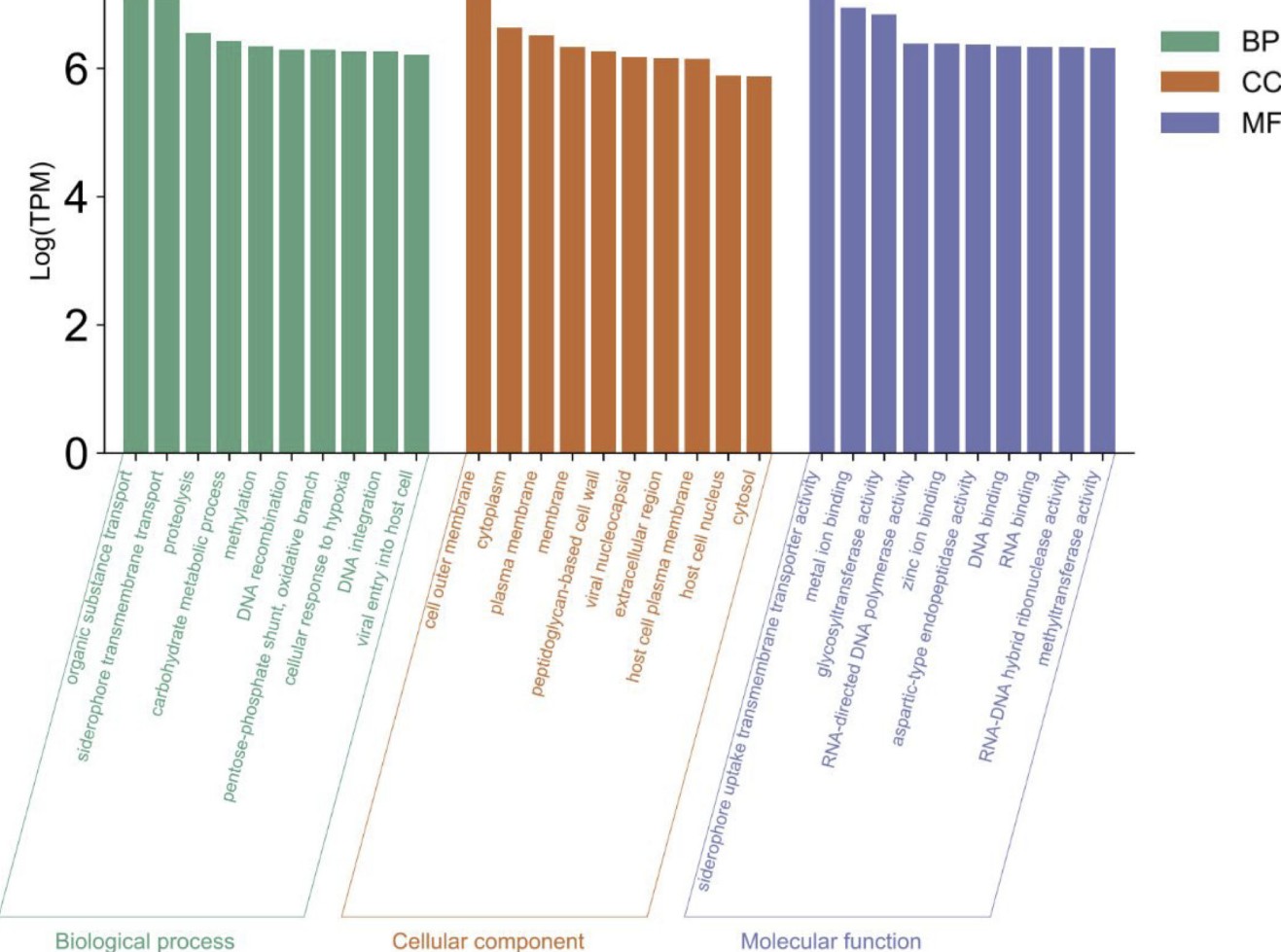

**FIG 5** GO analysis of microbial gene expression: GO term expression for BP, CC, and MF. Only the top 10 expressed GO terms are shown.

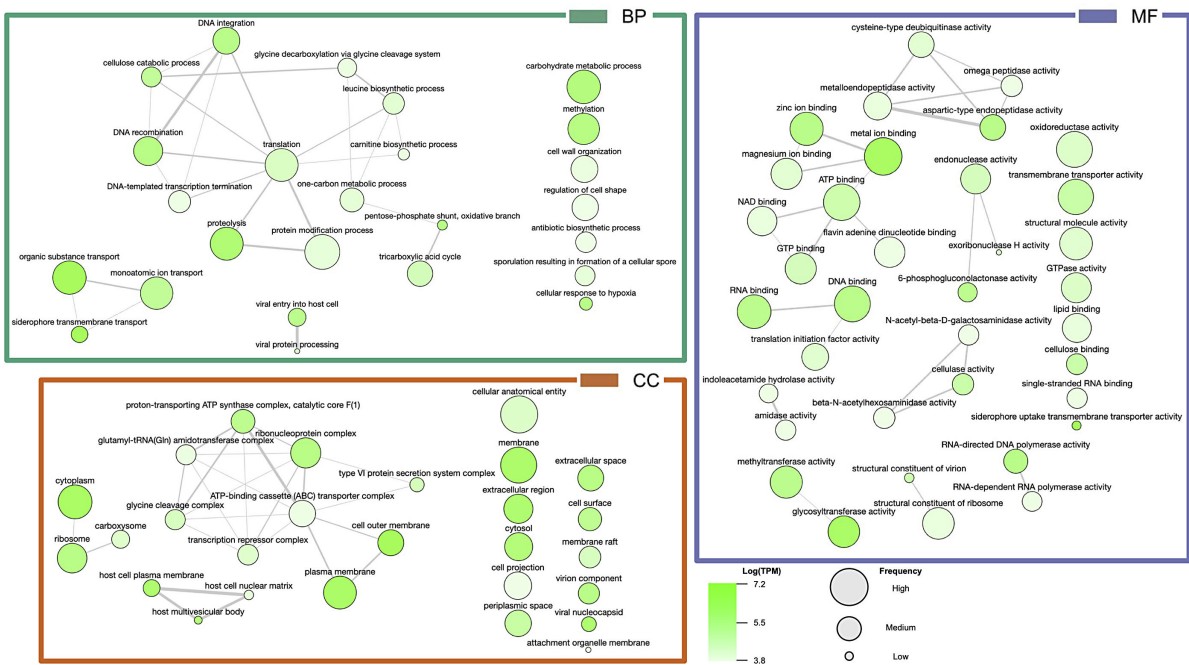

**FIG 6** Top 40 expressed bacterial GO terms from each category: BP, MF, and CC. Color of the bubble indicates Log(TPM) with dark-green being higher expression and light-green being lower expression. Bubble size indicates the frequency of the GO term in the underlying GOA database.

GO terms related to potential symbiotic relationships between the host and microbiome were identified and found to be highly expressed (Fig. 6). To understand the proteins involved with these processes, further filtering of gene expression data based on GO terms listed above as well as keywords such as "symbiont," "host," "antibiotic," and "DMSP" revealed gene expression involved in specific pathways and interactions of interest, such as DMSP degradation (Table S5), host interactions and symbiosis (Table S6), and antibiotic biosynthesis (Table S7). Gene expression involved with DMSP degradation revealed DMSP transporter proteins (OusV and OusW) and key enzymes involved with the production and regulation of acrylate through the DMSP cleavage pathway (DddP, PrpE, AcuI, 3-hydroxypropionyl-coenzyme A dehydratase, and Acryloyl-CoA reductase) (Table S5). Enzymes involved with the demethylation of DMSP were also found such as DmdB and DmdC, but no gene expression was found to be associated with DmdA or the production of methylmercaptopropionate (MMPA) from DMSP (Table S5). The proposed DMSP degradation pathway is shown in Fig. 7.

Genes potentially involved with host interactions and symbiosis fit into three subcategories "inhibit host immune system," "communication with host," and "communication between microbes" (Table S6). There were highly expressed genes associated with the "inhibit host immune system" such as effector proteins that alter host cell physiology and promote bacterial survival in host tissue (SspH2 and SlrP) and immune suppression or evasion of phagosomes, antiviral defenses, toxic metabolites, and antibiotics (PPE54, ICP0, and mycothiol S-conjugate amidase) (Table S6). Genes involved with "communication with host" included avirulence factors (AvrBS3, PthXo1, and AvrXa10), host immune system activation and protection against lethal gene expression and secondary infection (PPE34 and rexA), and symbiotic nodule development [NAD-ME, K(+)/H(+) antiporter subunit A/B and subunit D, BraC3, DctA2, and PpdK] (Table S6) which has also been shown in other marine invertebrates (42). Genes involved with "communication between microbes" included quorum-sensing genes and regulators involved with the suppression of bacterial growth [TqsA, LuxS, CdiA, CdiA2, pvdQ, and fill (Mhar_0446)] and sensory adaptation using methylation (PctA and TlpQ) (Table S6). Genes associated with the GO term "antibiotic biosynthesis process" (GO:0017000) were filtered for potential antibiotic

**FIG 7** Proposed DMSP cleavage pathway. Adopted from reference (41). DddP, Alma, PrpE, AcuI, and AcrC genes were identified in the microbial transcriptome (Table S5).

production (Table S7). Genes involved with antibiotic biosynthesis such as polyketides (PksJ, ThaG, and NcsB), etamycin (P4H), midecamycin (MdmC and MycG), rebeccamycin (RebG), pyrrolnitrin (PrnA and CpoP), gramicidin (LgrC, LgrB, and LgrD), oxytetracycline, (OxyS and OxyE), and plipastatin (PpsE and PpsB) were found in the microbial transcriptome (Table S7).

## DISCUSSION

### Location comparison and core microbiome

No statistical significance was found when comparing bacterial community beta diversity between populations or clustering into North and South groups (Fig. 4B). Also,

few differentially expressed genes were found between North and South populations (Table S4; Fig. 4A). These results support the presence of a uniform microbiome function and composition associated with the perivitelline fluid (PVF) of Kellet's whelk (Fig. 8). Although the composition was similar across all samples, the abundance values were highly variable, even within locations (Fig. S8). Any microbiome differences between populations of Kellet's whelk may be more of an indication of micro-environmental differences rather than that of a North vs. South boundary. Because our experiment was conducted in a common garden environment with over a year of acclimation for parents, there may be differences in the wild that are more dependent on the location and environment than parental origin. For example, a study on the microbiome of the sponge, *Ircinia campana*, was found to be both dependent on the environment as well as the host genotype (11). The microbiome diversity was significantly correlated with

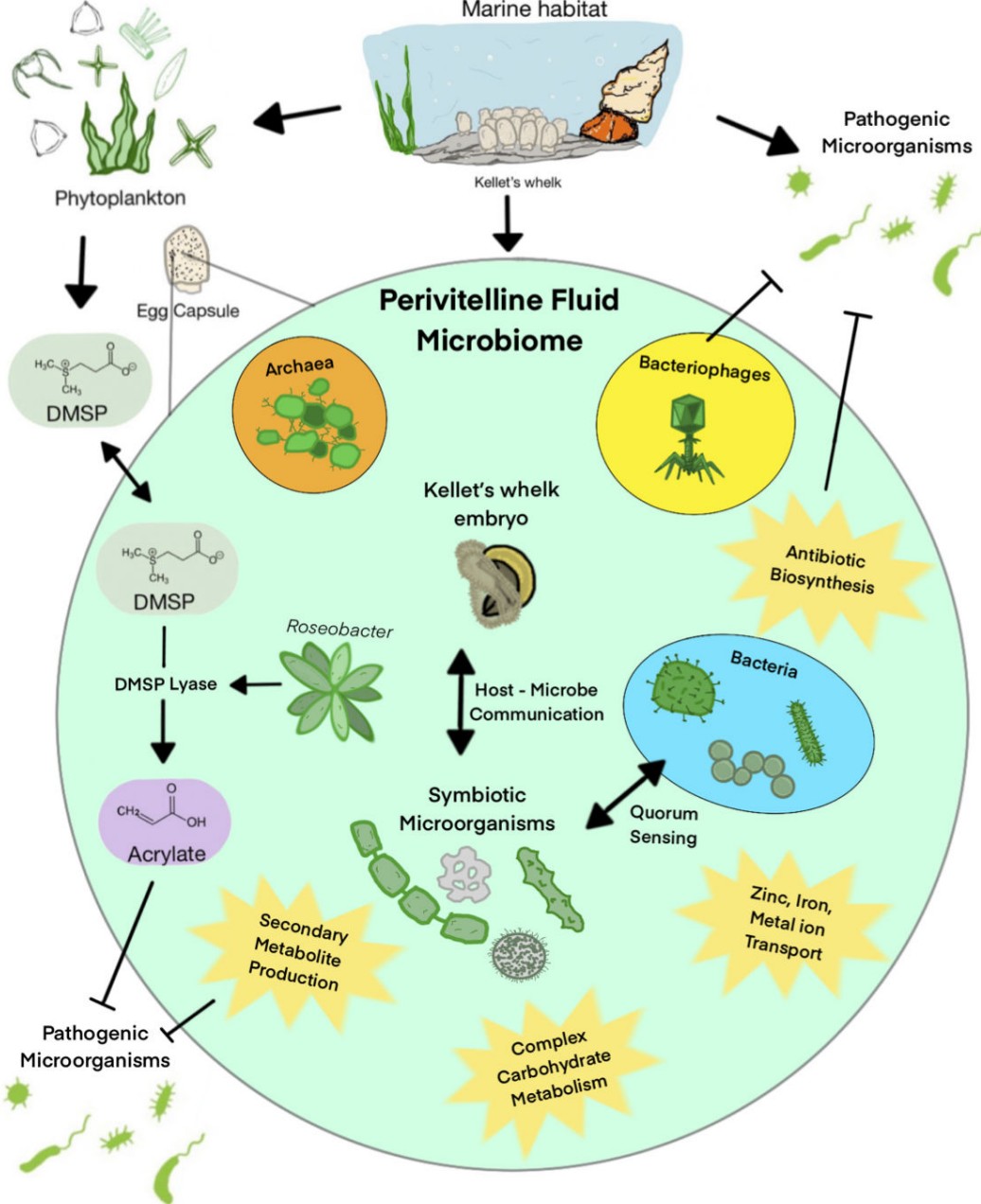

**FIG 8** Proposed function and defense mechanisms of Kellet's whelk PVF microbiome.

the genotype when comparing inter-location samples and was also correlated with the geographic location when comparing samples of the same genotype. By essentially eliminating the impact of diverse environmental conditions in various geographical locations, our experiment revealed a negligible or minimal correlation between geographic origin and bacterial composition.

## Archaeal and viral function

The few archaea species identified may have symbiotic relationships with the microbiome and host including *Candidatus giganthauma insulaporcus* associations with Gammaproteobacteria (43) and *Nitrosopumilus* spp. nitrifying ammonia to nitrites (Fig. S3) (44). The GO term analysis of gene expression of archaea revealed GO terms associated with processes such as "organonitrogen compound metabolic processes" and "ammonium transmembrane transporter activity" in support of processes above, although very few transcripts were identified indicating community expression was under-sampled from this group (Fig S7). Evidence of ammonia-oxidizing archaea species within the PVF suggests a potential role in detoxifying ammonia buildup, similar to their function in marine sponges and other organisms (45, 46). Further investigation of the archaeal community may elucidate symbiotic relationships in the PVF of marine invertebrates.

The abundant expression of bacteriophage-associated genes within the PVF suggests that the mucus-filled PVF may create a favorable environment for symbiotic viruses that can infect and eliminate pathogenic bacteria, thereby safeguarding the developing embryo (47). In fact, bacteriophages have been found to adhere to glycoproteins within the mucus of metazoan hosts where they are more likely to encounter bacteria before infection, providing an acquired antimicrobial immunity to the host (48). We propose some or all of the bacteriophage gene expression identified may be a signature of symbiosis within the PVF mucus (Table S8). These viral habitants may also be contributing to phage-mediated horizontal gene transfer that may be driving complex ecological interactions (49).

## Bacterial microbiome function

Based on the ranked top expressed GO terms, there is evidence of symbiotic relationships between the microbiome of Kellet's whelk PVF and Kellet's whelk embryos (Fig. 5). GO terms "siderophore transmembrane transport," "metal ion binding," "zinc ion binding," and "host cell plasma membrane" all indicate the maintenance and activity of a healthy microbiome often found in animal gut tissue (50–52). Top expressed GO terms mapped based on similarity shown in Fig. 6 such as the "carbohydrate metabolic process," "cellulose catabolic process," and "antibiotic biosynthesis" indicate potential functional aspects of the microbiome to Kellet's whelk host (Fig. 8). Microbiomes found within gut tissue often provide enzymes that can break down otherwise indigestible carbohydrates (53, 54). Moreover, gene expression related to antibiotic biosynthesis presents a direct defense provided by the microbiome against pathogenic bacteria and fungi and the potential for discovering new antibiotics (55, 56).

## Microbiome anti-microbial properties

Many microbe-host interactions provide protection to one another—in the instance of the Kellet's whelk egg capsule, the capsule membrane may act as a barrier for pathogenic organisms or interactions with the open ocean and encases a hospitable micro-environment for the development of Kellet's whelk embryos as well as symbiotic microorganisms. We believe these relationships are key to the development and prevention of pathogenic infection of Kellet's whelk embryos during a very vulnerable lifestage period. From our core microbiome analysis of Kellet's whelk PVF, we found a series of organisms that may provide a defensive chemical barrier while maintaining an environment ideal for the development of Kellet's whelk.

The most abundant bacteria (27.5%) found in our core microbiome were of the *Roseobacter* genus. This genus is known most broadly for its capability to metabolize DMSP (57, 58), create secondary metabolites, and facilitate energy acquisition by sulfur oxidation (59–61). This genus has been found to dominate communities in environments with high DMSP concentrations (62), and organisms within this genus may contain DMSP lyase (DddP), an enzyme capable of cleaving DMSP into acrylate and dimethylsulfide (DMS) (Fig. 7). Close homologs of this gene have been found in many *Roseobacter* species and some fungal species in which DMS production was shown (63, 64). Upon transcriptome inspection of Kellet's whelk PVF, a homolog to the DddP enzyme was found in association with the *Roseobacter* spp. within the microbiome as well as many other proteins associated with DMSP metabolism and regulation (Table S5). Upregulation of DMS and acrylate was found to be associated with a defensive role associated with clonal *Phaeocystis globosa* (65), and acrylate has been shown to directly inhibit the growth of bacterioplankton when at high concentrations in a closed system (66). Acrylate has also been shown to provide antimicrobial properties in the gut of pygoscelid penguins by the suppression of *Escherichia coli* populations and even an increase in growth rate of chicks at certain concentrations (0.01%–1%), but anorexia and death at concentrations above 10% (67). This suggests an intricate balance maintained by the microbiome within these marine animals to provide antimicrobial properties toward pathogenic microbes and other pathways that may be key to development. There may be pathways to both create and metabolize acrylate in order to maintain a homeostatic concentration that improves growth of Kellet's whelk embryos and protects them from infection. We have found expression of both acrylate-producing enzymes (DddP, *Rosebacter* spp.) and acrylate-digesting enzymes (PrpE, *Planktomarina* spp.) within the PVF (Table S5). We also did not find any gene expression of enzymes capable of degrading DMSP into other products such as MMPA, suggesting a direct degradation into acrylate and DMS. We suggest a potential pathway based on observed gene expression within the egg capsule PVF that may generate this system (Fig. 7).

*Roseobacter* spp. are also known for producing bacteriolytic agents against a wide range of fish and invertebrate pathogens (68) and many antimicrobial metabolites, such as tropodithietic acid (TDA) (69) and indigoidine, a blue pigment with antimicrobial properties, which has demonstrated the ability to inhibit marine bacteria (70, 71). *Roseobacter* spp. were discovered in the accessory nidamental gland in female Hawaiian bobtail squids (*Euprymna scolopes*) that the mother deposits into the PVF-like jelly that coats her eggs and suspected to protect the eggs from fouling, through antimicrobial compounds (25, 28, 71). Given *Roseobacter's* high abundance and diversity within Kellet's whelk PVF (Fig. 2), as well as its antimicrobial pathways and suspected involvement in symbiotic relationships (2, 70), we predict that the *Roseobacter* genus may significantly contribute to the protection of Kellet's whelk embryos during development through the metabolism of DMSP and production of defensive metabolites.

Similar to *Roseobacter* spp., *Sulfitobacter* spp., the third most abundant genus within Kellet's whelk PVF core microbiome, have also been found to produce secondary metabolites that may have antibacterial, antitumoral, and/or antiviral properties (3, 72–74).

As identified in the most expressed GO terms (Fig. 6), there were strong indications of antibiotic biosynthesis occurring from the microbiome gene expression. These genes were found to pertain to a variety of antibiotics produced by different bacteria. Some of these gene families were nearly complete with supportive BLAST scores for homology (Table S7). For example, Gramicidin is produced by four non-ribosomal peptide synthetases (i.e., LgrABCD) (75). We detected transcripts encoding LgrBCD, but not LgrA. Other antibiotic-producing genes had varying percent identity and bit-score suggesting homology but potentially a different biosynthesis pathway or product (Table S7). With so many different genes involved with antibiotic biosynthesis and varying homology, in conjunction with the fact that all reports here are novel discoveries for Kellet's whelk PVF, there may be new pathways for the production of unidentified antibiotics. The utilization

of homology-guided metagenome mining represents a highly effective approach for the identification of biologically significant natural products with medical relevance. Consequently, it encourages further investigation into the PVF meta-transcriptome for potential discoveries (76).

## Microbiome communication

Genes involved with host interactions and symbiosis were found to be involved with the inhibition of the host immune system, communication with the host, and communication between microbes (Table S6). Genes involved with the suppression and evasion of the host immune system suggest some of these microbes, potentially pathogenic or symbiotic, are able to inhibit the host immune system in order to maintain a hospitable environment to either themselves or all microbes within the PVF. For example, a gene involved with the detoxification of alkylating agents and antibiotics was found in Kellet's whelk PVF (mycothiol S-conjugate amidase) and may contribute to the survival of symbiont or pathogenic bacteria (Table S6) (77). Further investigation and validation of these findings will help elucidate their function in the PVF microbiome.

Genes involved with the communication with the host may be responsible for similar host immune system suppression as well as the induction of symbiotic relationships (i.e., PPE family protein PPE34 and avirulence proteins) (Table S6). Interestingly, there were an abundance of genes involved with symbiotic nodule development commonly found in plant symbiosis (Table S6). This was recently discovered in other marine invertebrate symbionts such as the lucinid bivalve (*Loripes lucinalis*) and the stilbonematid nematode (*Laxus oneistus*) and may be involved with nitrogen cycling (42).

Genes relating to the communication between microbes included genes involved with quorum sensing—a method in which microbial co-habitants communicate using pheromones to regulate gene expression (78). For example, transcripts associated with autoinducer 2 (AI-2) synthesis and transport were discovered in the PVF, suggesting that members of the microbiome can participate in inter- and intra-cellular cross talk (Table S6) (79). This particular quorum-sensing molecule has been found to regulate many different metabolic pathways including biofilm formation, nitrogen fixation, and even antibiotic peptide regulation (80–82) and, therefore, exhibits the potential to modulate one or multiple pathways involved in Kellet's whelk PVF. Other molecules were found to control the growth of adjacent bacteria in a contact-dependent fashion (CdiA and CdiA2) (83), and sensory adaptation through chemotactic transducer genes that respond to environmental stimuli modify methyl-groups and, in turn, the regulation of different metabolic pathways (PctA and TlpQ) (84). These chemoreceptors have also been found to bind to AI-2 in order to induce biofilm formation (80).

This study represents a comprehensive analysis of a biological system to understand both what constitutes Kellet's whelk PVF microbiome and how the microbiome varies with location origin, plus what function this microbiome may offer during this vulnerable life history stage. As illustrated in Fig. 8, we present the potential functions of this PVF microbiome as supported by 16S sequencing and gene expression analysis.

## Limitations and future direction

There may be limitations to our RNA-seq analysis due to the RNA-seq libraries being prepared from total RNA using poly(A) enrichment of the mRNA (mRNA-seq). Therefore, not all microbes and their gene expression may be represented in the microbial transcriptome analysis. Yet, there were still significant and consistent results obtained through the microbial transcriptome assembly using BLAST, where all microbial gene expression was identified based on having a higher BLAST bitscore than any eukaryotic gene from the UniProt database (high confidence in these genes originating from microbial members). This study is a great example of leveraging data produced for other studies to gain insight into novel research topics.

To gain greater insight into the microbial differences among Kellet's whelk populations, we suggest integrating 16S sequencing, total RNA-seq, and protein extraction

and identification of PVF from locally collected samples from all four locations (NAP, MON, POL, and DIC) and comparing to that of the core microbiome composition and transcriptome activity identified in this study. This study used samples with parents kept in a common garden crossing study for a year prior to analysis of the microbiome; therefore, any differences in the microbiomes across locations would be a result of retained habitants through this process. Using locally collected samples and seawater samples directly from each location in combination with what we have found in this study could help shed light on the micro-environments experienced by different populations and the function and composition of microorganisms specific to local environments. This may also increase our understanding of this study's results in relation to whether the most determining factor of the PVF microbiome is the mother's origin or the most recent environmental conditions. Our comparison of the microbiome between locations assumes that the microbiomes of each egg capsule were directly transferred from the mother rather than the current environment. In order to understand the origin of the PVF microbiome, more extensive research regarding the maternal albumen gland microbiome may elucidate new pathways beyond the various squid ANG systems for the vertical transference of beneficial microbes to invertebrate eggs (29).

## ACKNOWLEDGMENTS

This material is based upon work supported by the National Science Foundation under grant number OCE-1924537 and the Frost Summer Undergraduate Research Program through the Bailey College of Science and Mathematics at California Polytechnic State University, San Luis Obispo.

The bioinformatics analysis was carried out through a server at California Polytechnic State University operated by the Bioinformatics Research Group (BIRG) with special thanks to Dr. Paul Anderson.

Sample preparation and extractions were conducted at the Center for Applied Biotechnology at California Polytechnic State University.

The 16S rRNA gene sequencing was carried out by MR DNA Shallowater, Texas, USA.

The RNA sequencing was carried out by Novogene at the University of California, Davis, USA.

## AUTHOR AFFILIATION

[1]Department of Biological Sciences, California Polytechnic State University, San Luis Obispo, California, USA

## AUTHOR ORCIDs

Benjamin N. Daniels http://orcid.org/0009-0002-1202-0765
Pat Fidopiastis http://orcid.org/0000-0003-1631-4744

## FUNDING

| Funder | Grant(s) | Author(s) |
| --- | --- | --- |
| National Science Foundation (NSF) | OCE-1924537 | Crow White |
| FROST Research Fund, California Polytechnic State University | | Jenna Nurge |

## AUTHOR CONTRIBUTIONS

Benjamin N. Daniels, Conceptualization, Data curation, Formal analysis, Funding acquisition, Investigation, Methodology, Project administration, Resources, Software, Supervision, Validation, Visualization, Writing – original draft, Writing – review and editing | Jenna Nurge, Conceptualization, Data curation, Formal analysis, Funding acquisition, Investigation, Methodology, Visualization, Writing – original draft, Writing – review and editing | Chanel De Smet, Conceptualization, Data curation, Formal

analysis, Investigation, Writing – original draft, Writing – review and editing | Olivia Sleeper, Conceptualization, Data curation, Formal analysis, Investigation, Methodology, Writing – original draft, Writing – review and editing | Crow White, Conceptualization, Funding acquisition, Methodology, Project administration, Resources, Supervision, Writing – review and editing | Jean M. Davidson, Conceptualization, Funding acquisition, Methodology, Project administration, Resources, Supervision, Writing – review and editing | Pat Fidopiastis, Conceptualization, Funding acquisition, Investigation, Methodology, Project administration, Resources, Supervision, Writing – review and editing

## DATA AVAILABILITY

All 16S sequencing data results produced by MR DNA are available at https://drive.google.com/drive/folders/1m5Te_38rqFN8nwo1JytAk43kRrhnMWpa. All RNA-seq data have been uploaded to the Sequence Read Archive (SRA) under the NCBI BioProject number PRJNA1000198. The microbiome transcriptome has been uploaded to GitHub and is available at https://github.com/bndaniel/Kellets-whelk-microbiome transcriptome/blob/main/microbiome_transcriptome.fasta. All commands and R scripts used can be found on Github.

## ADDITIONAL FILES

The following material is available online.

### Supplemental Material

**Fig. S1-S9 (Spectrum03514-23-s0001.pdf).** Supplementary figures.
**Tables S1-S9 (Spectrum03514-23-s0002.pdf).** Supplementary tables.

### Open Peer Review

**PEER REVIEW HISTORY (review-history.pdf).** An accounting of the reviewer comments and feedback.

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
