## [Reviewer comments · Microbiology Spectrum]

Microbiology Spectrum

Microbiome Composition and Function Within the Kellet's Whelk Perivitelline Fluid

Benjamin Daniels, Jenna Nurge, Chanel De Smet, Olivia Sleeper, Crow White, Jean Davidson, and Pat Fidopiastis

Corresponding Author(s): Benjamin Daniels, California Polytechnic State University Biological Sciences Department

Review Timeline:

Submission Date:	October 2, 2023
Editorial Decision:	November 20, 2023
Revision Received:	January 16, 2024
Accepted:	January 17, 2024

Editor: Erik Hom

Reviewer(s): The reviewers have opted to remain anonymous.

Transaction Report:

DOI: <https://doi.org/10.1128/spectrum.03514-23>

Re: Spectrum03514-23 (Microbiome Composition and Function Within the Kellet's Whelk Perivitelline Fluid)

Dear Dr. Benjamin Norha Daniels:

Thank you for the privilege of reviewing your work and for your patience. Your manuscript has now been reviewed by two experts. Below you find reviewers' comments and instructions on what to include in your revised submission.

Both reviewers were quite supportive of publishing your work pending some revisions. Please tone down some of the statements you make in the conclusions, especially with regards to functions like immune suppression and nitrogen fixation as it's not clear your data definitively support those conclusions. Please include more background information about the biology of the host and how the pvf microbiome may interact with the embryo. Lastly, as suggested by Reviewer #2, please refine and more clearly articulate the main points of your work.

Revision Guidelines

- In your cover letter, please summarize for me the changes you have made in your revised manuscript.
- Upload point-by-point responses to the issues raised by the reviewers in a file named "Response to Reviewers," NOT IN YOUR COVER LETTER
- Upload a compare copy of the manuscript (without figures) as a "Marked-Up Manuscript" file
- Upload a clean .DOC/.DOCX version of the revised manuscript and remove the previous version
- Each figure must be uploaded as a separate, editable, high-resolution file (TIFF or EPS preferred), and any multipanel figures must be assembled into one file
- Any supplemental material intended for posting by ASM should be uploaded separate from the main manuscript; you can combine all supplemental material into one file (preferred) or split it into a maximum of 10 files, with all associated legends included

Sincerely,
Erik Hom
Editor
Microbiology Spectrum

Reviewer #1 (Public repository details (Required)):

All data have been deposited in the SRA or made publicly available.

Reviewer #1 (Comments for the Author):

This study characterizes the microbial community and gene expression of the perivitelline fluid (PVF) found in Kelleys Whelk eggs obtained from laboratory-maintained animals that were originally collected from four locations along the California coast. Egg-associated microbiomes can have a variety of protective functions for their hosts, especially in aquatic/marine environments. This study is the first to characterize the microbiome of the PVF from an invertebrate while also analyzing microbial gene expression to better understand possible functions in this association. The identification of the microbiome using long read sequencing combined with RNAseq provides interesting insight on this understudied association. The authors did a great deal of work in their analyses and is an important contribution to the field. The manuscript is well written and will appeal to the broad readership of *Microbiology Spectrum*. However, there are some points that should be addressed.

Specific comments:

Some additional background on the development and basic biology of Kelleys Whelk would be helpful for readers to understand possible interactions between the perivitelline fluid microbiome and the host. Specifically, how long is embryogenesis (weeks? months)? I assume at least that since some samples were collected at 35-45 days into embryogenesis. More information about the life cycle would give valuable context about the possible function of the microbiota in the perivitelline fluid and interactions with the embryo, especially given the functional hypotheses proposed by the authors in the discussion.

Line 66: As an updated reference for Collins 2015 there is now more known about how bacteria associated with the jelly coat (similar to PVF) in eggs of the Hawaiian bobtail squid protect against specific fungal pathogens and data that show that the microbiome in the egg jelly is protective (Kerwin et al. 2019 *mBio*; PMID: 31662458)

Lines 77-79 The authors state that the microbiome may be transmitted from the mother to the PVF. Is there any direct evidence for this and how may that work biologically? If vertically transmitted from the mother, are these bacteria stored in a specialized organ or associated with the oviduct? Do the authors hypothesize that these bacteria are transferred during egg deposition? Further discussion of potential modes of transmission either in the introduction or discussion would help provide context about this association.

Lines 84-95. Some of these details are more appropriate for the methods and could be edited down in the introduction.

Lines 109-110 (as well as results and discussion). Since the whelks were adapted to a laboratory setting over a period of a year before eggs were sampled, do the authors think this may have influenced their results esp. in the context of geography on similarities they observed in the PVF microbiomes? This also gets back to the point I brought up by vertical transmission. From lines 136-138, there seems to be an assumption that if the bacteria are vertically transmitted that the PVF microbiome of the mother may correlate with geography but since the mothers weren't sampled in this study I think it's difficult to make that point and this could be brought up as a future direction in the discussion. The results are still very interesting and the authors found differences in the egg PVF that came from parents collected from different locations, but I would also highlight that future work should characterize the microbiome of the parent whelks and/or eggs (if possible directly from the field) along with environmental samples. Some of these points were mentioned in the discussion line 398-400 and in the limitations and future directions section but could be expanded.

Line 147: Would add "DNA extractions from 4 egg capsules from laboratory-maintained animals that were originally collected at each location..." or something similar to make it clear that the egg capsules were not collected from the environment at each location.

Line 178: suggest defining what you mean by extremely accurate (to species or strain level?)

Line 191: Was RNA extracted from just the PVF or whole egg with embryo? It appears from the section starting in line 207 that host embryonic RNA also extracted? If so, what percentage of the reads were host vs. microbial? This could be reported in a supplementary table as I didn't see this information in the methods or results.

Materials and methods: did the authors sample the tanks or local seawater where the whelks and eggs were kept? This could serve as an important control for understanding how the local environment may influence the PVF microbiome. If not, this could be suggested as a future direction in the discussion.

Line 266: We to we

Lines 324: The text says there were 301 total viral genes but Table 2 shows 2,281.

Section starting 335 Microbiome Function: I recommend that some of this section be edited down and focus on just the most

abundant GO terms and functions. Much of this section lists terms that are shown in the figures or tables.

Fig. 6 Line 387 can the authors add a scale for the log (TPM) and frequencies to better show quantitative differences?

Line 408 (section Archaeal and viral function): Since such few transcripts were identified for these groups, especially archaea (only 301 genes) I would be cautious about assigning function and would add a sentence that community expression was under sampled from this group.

Line 433 suggest Bacterial microbiome function for this header

Lines 491-493 Indigoidine derived from a roseobacter (*Leisingera* sp. JC1) has also been described from an egg bacterium of the Hawaiian bobtail squid (Gromek et al., 2016; PMID: 27660622)

Lines 502-503 the third most abundant genera...have to the third most abundant genus...has

Line 510-511 Please add a reference for this statement about gramicidin.

Line 524: I would tone this statement down. The data about suppression of the immune system is somewhat speculative at this point without including data for host gene expression. You can say that the data suggest that the PVF microbiome may influence the host immune system but this will require further experiments to show.

Lines 531-537 and Fig. 8: I would be cautious about inferring too much with regard to nitrogen fixation especially since the key bacterial nitrogenase gene doesn't appear to have been detected in your data set, unless I missed that. Fig. 8 also implies that archaea in the PVF undergo nitrogen fixation but I don't see evidence for this from the gene expression data. Perhaps a better term would ammonia oxidation or nitrifying processes.

Lines beginning 559 Limitations and future directions: I appreciated inclusion of this section and suggest adding a few more items including some of the points mentioned above. A couple of other suggestions for future directions might include experimentation on eggs obtained from the wild to alter the PVF microbiome experimentally (e.g. antibiotic or other treatment) and/or in vitro work with cultured microbiome members.

Reviewer #2 (Public repository details (Required)):

This study includes 16S rRNA amplicon sequencing data and RNAseq data. The authors have submitted these data to the NCBI SRA.

Reviewer #2 (Comments for the Author):

Daniels et al. provide a comprehensive overview of the microbiome of perivitelline fluid in *Kellet's* whelks. They show that whelks collected from different geographic locations show similar microbiome functionality when kept in a common garden for ~1 year. I thought the methodology of this paper - paired PacBio 16S amplicon sequencing with RNAseq - was clever and innovative and provides a approach more fit to functional microbiome study than standard metagenomics. I also like the concept of studying perivitelline fluid, which is a portion of the invertebrate life cycle that is very poorly characterized to this point. However, I have a few points that I believe would help the clarity of the article, outlined below.

Major points:

1. Though I very much enjoyed reading this article, it was a bit difficult to follow the thread from beginning to end. I would recommend choosing a central point and making sure your methods and results tie back to that point. For example, I found the most compelling theme to be "The functional microbiome of *Kellet's* whelk is similar across sites and provides some gene functions potentially beneficial to healthy growth and development", while the introduction of the separate Archaea sequencing and the time spent on DMSP pathways seem to not fully fit this theme until late in the discussion.

2. I may have missed it, but I did not see any controls mentioned in these analyses. Were extraction or seawater controls included in the analysis? I do not believe it would make a significant difference to the results, but some of the less abundant core taxa may be a result of contaminants which arise during most protocols of this fashion. If controls were run, please make sure they are presented clearly.

Minor points:

Line 46 - This sentence is a bit confusing, suggest rephrasing for clarity.

Lines 66-68 - I would recommend rephrasing to simply introduce the hypothesis. This makes it sound like your hypothesis has already been tested by others, reducing novelty.

Line 114 - I recommend changing to "across a portion of the Kellet's whelk biogeographic range" as no sampling was done in Baja, where they persist.

Line 118-119 - this is a bit confusing, what happened to the other samples?

Lines 100-125 - This whole paragraph could be removed to save space, as much of this is reiterated elsewhere in the methods. Be sure to keep the coordinates and permit number somewhere in the methods, though.

Line 142 - What is MON x NAP? Were there crosses done during the common garden? If so, these should be discussed further.

Line 159 - Suggest rephrasing to "35-cycle PCR protocol"

Line 160 - Suggests adding "Samples were cycled at 94C..."

Lines 167-169 - Which samples? Why did you do this?

Line 172 - What database is this? Who curated it? Can you provide a more specific link, as this one leads simply to the NCBI homepage?

Line 213 - UniProt

Line 218 - Were there any hits to Eukaryota that did not represent *K. kelletii*? Such as potential parasites, etc.?

Table 2 - I would suggest switching this Table with Table S4, which provides more detailed information that is of interest to the reader.

Line 355 - Why is DMSP degradation of interest? This needs some context.

Figure 5A caption - Suggest rephrasing to "differential expression of 22 genes"

Lines 393-395 - This sentence is self-contradictory. Maybe you could say the Kellet's whelk microbiome is relatively consistent in function, but not in taxonomic composition, or something similar.

Figure 8 - This figure is very well done. If it is possible to include a graphical abstract, this would help your readers with a lot of context early on.

Response to Reviewers

“Microbiome Composition and Function Within the Kelleys Whelk Perivitelline Fluid”

Benjamin N Daniels (corr-auth), Jenna Nurge, Chanel De Smet, Olivia Sleeper, Crow White, Jean M Davidson, Pat Michael Fidopiastis

We would like to thank the Editor and Reviewers for their time and comments, which provide constructive feedback and insight. A point-by-point response is given below with the Reviewer's comments in black and our responses in blue.

Editor's Decision

Both reviewers were quite supportive of publishing your work pending some revisions. Please tone down some of the statements you make in the conclusions, especially with regards to functions like immune suppression and nitrogen fixation as its not clear your data definitively support those conclusions.

We have toned down our statement in the conclusion regarding immune suppression and nitrogen fixation to focus on what is reflected by our results (See the discussion in the “tracked changes document”).

Please include more background information about the biology of the host and how the pvf microbiome may interact with the embryo.

We have included a greater background of PVF biology and the interactions with the embryo in lines 51-79.

Lastly, as suggested by Reviewer #2, please refine and more clearly articulate the main points of your work.

All authors have now revised the manuscript for clarity and to stay focused on articulating the main points of our work that are supported by the results.

Reviewer #1 (Comments for the Author):

This study characterizes the microbial community and gene expression of the perivitelline fluid (PVF) found in Kellet Whelk eggs obtained from laboratory-maintained animals that were originally collected from four locations along the California coast. Egg-associated microbiomes can have a variety of protective functions for their hosts, especially in aquatic/marine environments. This study is the first to characterize the microbiome of the PVF from an invertebrate while also analyzing microbial gene expression to better understand possible functions in this association. The identification of the microbiome using long read sequencing combined with RNAseq provides interesting insight on this understudied association. The authors did a great deal of work in their analyses and is an important contribution to the field. The manuscript is well written and will appeal to the broad readership of Microbiology Spectrum. However, there are some points that should be addressed.

(Public repository details (Required)):

All data have been deposited in the SRA or made publicly available.

Specific comments:

Some additional background on the development and basic biology of Kellet's Whelk would be helpful for readers to understand possible interactions between the perivitelline fluid microbiome and the host. Specifically, how long is embryogenesis (weeks? months)? I assume at least that since some samples were collected at 35-45 days into embryogenesis. More information about the life cycle would give valuable context about the possible function of the microbiota in the perivitelline fluid and interactions with the embryo, especially given the functional hypotheses proposed by the authors in the discussion.

We have included a greater background information on the biology of Kellet's whelk, including its life cycle (bi-partite) and embryogenesis period (approx 35-45 days), and possible interactions between the perivitelline fluid microbiome and the host, in lines 51-79.

Line 66: As an updated reference for Collins 2015 there is now more known about how bacteria associated with the jelly coat (similar to PVF) in eggs of the Hawaiian bobtail squid protect against specific fungal pathogens and data that show that the microbiome in the egg jelly is protective (Kerwin et al. 2019 mBio; PMID: 31662458)

Thank you for the updated reference, which we have integrated into the manuscript.

Lines 77-79 The authors state that the microbiome may be transmitted from the mother to the PVF. Is there any direct evidence for this and how may that work biologically? If vertically transmitted from the mother, are these bacteria stored in a specialized organ or associated with the oviduct? Do the authors hypothesize that these bacteria are transferred during egg deposition? Further discussion of potential modes of transmission either in the introduction or discussion would help provide context about this association.

Thank you for bringing this up. We have added a more extensive explanation, with additional references, of how we hypothesize this works biologically in lines 51-79. This hypothesis has also been added to our future directions paragraph lines 576-587, as it requires more extensive examination.

Lines 84-95. Some of these details are more appropriate for the methods and could be edited down in the introduction.

We have removed these details and added them to the methods. See lines 93-102.

Lines 109-110 (as well as results and discussion). Since the whelks were adapted to a laboratory setting over a period of a year before eggs were sampled, do the authors think this may have influenced their results esp. in the context of geography on similarities they observed in the PVF microbiomes? This also gets back to the point I brought up by vertical transmission. From lines 136-138, there seems to be an assumption that if the bacteria are vertically transmitted that the PVF microbiome of the mother may correlate with geography but since the mothers weren't sampled in this study I think it's difficult to make that point and this could be brought up as a future direction in the discussion. The results are still very interesting and the authors found differences in the egg PVF that came from parents collected from different locations, but I would also highlight that future work should characterize the microbiome of the parent whelks and/or eggs (if possible directly from the field) along with environmental samples. Some of these points were mentioned in the discussion line 398-400 and in the limitations and future directions section but could be expanded.

Thank you for bringing this to our attention. We actually think that the experimental design – acclimatization of mothers from different geographies in a common garden environment prior to the production and sampling of their PVF – is a strength, because it helps remove effects of environmental conditions, such as food, water and light, which are controlled in the experimental aquaria, on variability in the PVF microbiome. Because the environment is controlled for, variability in PVF can more confidently be attributed to retained microbial differences in the mothers associated with the geographies from which they were sampled. Said another way, microbiome differences that are maintained even after acclimation to a common garden are expected to

represent differences among mother origin, rather than just differing environments (Villemereuil et al. 2016 *Heredity* 116:249-254).

Nonetheless, we agree with the reviewer that analysis of egg capsules collected in the field could help elucidate differences in each location's microbiome; it would, however, add additional independent variables (geography and environment, rather than just geography), so extra care in the statistics would be important. This experimental design would be most powerful in coordination with our current experiment to de-couple what differences in the PVF microbiome may be a result of geographic location versus the environment. Since we did not see significant differences in the microbiome with geographic location, it is also possible that the PVF microbiome is heavily determined by the most current environmental conditions rather than the mother's origin.

We have added more discussion on this topic and to the section covering limitations of our study and future directions in lines 393-405 and 573-582.

Line 147: Would add "DNA extractions from 4 egg capsules from laboratory-maintained animals that were originally collected at each location..." or something similar to make it clear that the egg capsules were not collected from the environment at each location.

We have added this clarification.

Line 178: suggest defining what you mean by extremely accurate (to species or strain level?)

We have replaced "extremely accurate" with more descriptive language ("... accurate to the genus and species level").

Line 191: Was RNA extracted from just the PVF or whole egg with embryo? It appears from the section starting in line 207 that host embryonic RNA also extracted? If so, what percentage of the reads were host vs. microbial? This could be reported in a supplementary table as I didn't see this information in the methods or results.

4.19% of all genes identified and 10.39% of all RNA transcripts originated from the microbiome. We have added a statement addressing this in lines 337-338.

Materials and methods: did the authors sample the tanks or local seawater where the whelks and eggs were kept? This could serve as an important control for understanding how the local environment may influence the PVF microbiome. If not, this could be suggested as a future direction in the discussion.

Due to cost constraints, we focused our data collection on replicates rather than controls. We have added this to our future directions statement lines 576-584.

Line 266: We to we

Done.

Lines 324: The text says there were 301 total viral genes but Table 2 shows 2,281.

We have fixed this error.

Section starting 335 Microbiome Function: I recommend that some of this section be edited down and focus on just the most abundant GO terms and functions. Much of this section lists terms that are shown in the figures or tables.

We have thinned this section to focus on the most abundant GO terms and functions. See lines 333-383.

Fig. 6 Line 387 can the authors add a scale for the log (TPM) and frequencies to better show quantitative differences?

We have added a color scale with quantitative markers and frequency markers with descriptions. See Fig. 6.

Line 408 (section Archaeal and viral function): Since such few transcripts were identified for these groups, especially archaea (only 301 genes) I would be cautious about assigning function and would add a sentence that community expression was under sampled from this group.

Thank you for this input. We have added this to our discussion in lines 407-418 and toned down our interpretation of archaeal functional activity.

Line 433 suggest Bacterial microbiome function for this header

We have changed the header for this section to "Bacterial Microbiome Function".

Lines 491-493 Indigoidine derived from a roseobacter (*Leisingera* sp. JC1) has also been described from an egg bacterium of the Hawaiian bobtail squid (Gromek et al., 2016; PMID: 27660622)

Thank you for adding this reference. We have added it to our discussion lines 485-493.

Lines 502-503 the third most abundant genera...have to the third most abundant genus...has

We have fixed this error.

Line 510-511 Please add a reference for this statement about gramicidin.

We have added a citation to this statement.

Schracke N, Linne U, Mahlert C, Marahiel MA (June 2005). "Synthesis of linear gramicidin requires the cooperation of two independent reductases". *Biochemistry*. 44 (23): 8507–13. doi:10.1021/bi050074t

Line 524: I would tone this statement down. The data about suppression of the immune system is somewhat speculative at this point without including data for host gene expression. You can say that the data suggest that the PVF microbiome may influence the host immune system but this will require further experiments to show.

We have toned down this statement and rephrased it as speculative. See lines 519-529.

Lines 531-537 and Fig. 8: I would be cautious about inferring too much with regard to nitrogen fixation especially since the key bacterial nitrogenase gene doesn't appear to have been detected in your data set, unless I missed that. Fig. 8 also implies that archaea in the PVF undergo nitrogen fixation but I don't see evidence for this from the gene expression data. Perhaps a better term would ammonia oxidation or nitrifying processes.

Upon further review of this section, we have removed our interpretation of nitrogen fixation and generalized to potential nitrogen cycling. See lines 530-536.

Lines beginning 559 Limitations and future directions: I appreciated inclusion of this section and suggest adding a few more items including some of the points mentioned above. A couple of other suggestions for future directions might include experimentation

on eggs obtained from the wild to alter the PVF microbiome experimentally (e.g. antibiotic or other treatment) and/or in vitro work with cultured microbiome members.

We have added the recommended limitations and future directions. See lines 569-587. Thank you!

Reviewer #2 (Comments for the Author):

Daniels et al. provide a comprehensive overview of the microbiome of perivitelline fluid in Kellet's whelks. They show that whelks collected from different geographic locations show similar microbiome functionality when kept in a common garden for ~1 year. I thought the methodology of this paper - paired PacBio 16S amplicon sequencing with RNAseq - was clever and innovative and provides a approach more fit to functional microbiome study than standard metagenomics. I also like the concept of studying perivitelline fluid, which is a portion of the invertebrate life cycle that is very poorly characterized to this point. However, I have a few points that I believe would help the clarity of the article, outlined below.

Data repository:

This study includes 16S rRNA amplicon sequencing data and RNAseq data. The authors have submitted these data to the NCBI SRA.

Major points:

1. Though I very much enjoyed reading this article, it was a bit difficult to follow the thread from beginning to end. I would recommend choosing a central point and making sure your methods and results tie back to that point. For example, I found the most compelling theme to be "The functional microbiome of Kellet's whelk is similar across sites and provides some gene functions potentially beneficial to healthy growth and development", while the introduction of the separate Archaea sequencing and the time spent on DMSP pathways seem to not fully fit this theme until late in the discussion.

Thank you for bringing this to our attention. We have revised the narrative of the manuscript to focus on the central theme of the study (as pointed out by the reviewer), that the PVF microbiome is not significantly different between geographies and it provides gene functions that may be beneficial to egg growth and development. We have shifted our focus away from the archaea sequencing and clarified the focus on the DMSP pathway. We have made changes as shown in the "tracked changes document".

2. I may have missed it, but I did not see any controls mentioned in these analyses. Were extraction or seawater controls included in the analysis? I do not believe it would make a significant difference to the results, but some of the less abundant core taxa may be a result of contaminants which arise during most protocols of this fashion. If controls were run, please make sure they are presented clearly.

Due to cost constraints, we focused our data collection on replicates rather than controls. We have added this as a limitation of our study. See lines 284-286.

Minor points:

Line 46 - This sentence is a bit confusing, suggest rephrasing for clarity.

This line has been rephrased for clarity.

Lines 66-68 - I would recommend rephrasing to simply introduce the hypothesis. This makes it sound like your hypothesis has already been tested by others, reducing novelty.

Thank you for this comment. We have rephrased to remove indication that our hypothesis has already been tested.

Line 114 - I recommend changing to "across a portion of the Kellet's whelk biogeographic range" as no sampling was done in Baja, where they persist.

We have clarified this with a "portion" rather than the entire biogeographic range.

Line 118-119 - this is a bit confusing, what happened to the other samples?

This line has been rephrased for clarity.

Lines 100-125 - This whole paragraph could be removed to save space, as much of this is reiterated elsewhere in the methods. Be sure to keep the coordinates and permit number somewhere in the methods, though.

We have removed this paragraph and consolidated this information into the methods.

Line 142 - What is MON x NAP? Were there crosses done during the common garden? If so, these should be discussed further.

MON x NAP represent potential crosses conducted during the common garden. These samples were only used for total microbiome transcriptomics and not for comparative analysis of locations. We have elaborated on this further in line 132.

Line 159 - Suggest rephrasing to "35-cycle PCR protocol"

Done.

Line 160 - Suggest adding "Samples were cycled at 94C..."

We have fixed this error.

Lines 167-169 - Which samples? Why did you do this?

These samples were sequenced for archaeal diversity and abundance. We have added clarification see lines 161-164.

Line 172 - What database is this? Who curated it? Can you provide a more specific link, as this one leads simply to the NCBI homepage?

This database was regularly updated and curated by MR DNA. We have added clarity to this statement.

Line 213 - UniProt

This error has been fixed.

Line 218 - Were there any hits to Eukaryota that did not represent *K. kellestia*? Such as potential parasites, etc.?

We did not focus on these hits (they were filtered out with all eukaryotic hits) as they would have required more extensive filtering and greater knowledge of potential eukaryotic parasites.

Table 2 - I would suggest switching this Table with Table S4, which provides more detailed information that is of interest to the reader.

We have swapped these tables.

Line 355 - Why is DMSP degradation of interest? This needs some context.

DMSP degradation was introduced as a means for producing antimicrobial metabolites. We have added more context and additional references for this analysis to the introduction in lines 39-43.

Figure 5A caption - Suggest rephrasing to "differential expression of 22 genes"

We have rephrased as recommended.

Lines 393-395 - This sentence is self-contradictory. Maybe you could say the Kellet's whelk microbiome is relatively consistent in function, but not in taxonomic composition, or something similar.

We have rephrased this sentence for clarity. See lines 388-393.

Figure 8 - This figure is very well done. If it is possible to include a graphical abstract, this would help your readers with a lot of context early on.

Thank you for your comments! We have requested to include this figure as a graphical abstract.

Re: Spectrum03514-23R1 (Microbiome Composition and Function Within the Kellet's Whelk Perivitelline Fluid)

Dear Dr. Benjamin Norha Daniels:

Happy New Year and thanks for your revisions. Your manuscript has been accepted, and I am forwarding it to the ASM production staff for publication. Your paper will first be checked to make sure all elements meet the technical requirements. ASM staff will contact you if anything needs to be revised before copyediting and production can begin. Otherwise, you will be notified when your proofs are ready to be viewed.

Sincerely,
Erik Hom
Editor
Microbiology Spectrum